# Maternal haemodynamics in Hypertensive Disorders of Pregnancy under antihypertensive therapy (HyperDiP): study protocol for a prospective observational case–control study

Pilar Palmrich ,[1] Nadine Haase,[2,3,4,5] Meryam Sugulle,[6,7] Erkan Kalafat,[8] Asma Khalil,[9] Julia Binder[1]

For numbered affiliations see end of article.

**Correspondence to**
Dr Julia Binder;
julia.binder@meduniwien.ac.at

## ABSTRACT

**Introduction** Hypertensive disorders of pregnancy (HDP) are associated with a high incidence of maternal and perinatal morbidity and mortality. HDP, in particular pre-eclampsia, have been determined as risk factors for future cardiovascular disease. Recently, the common hypothesis of pre-eclampsia being a placental disorder was challenged as numerous studies show evidence for short-term and long-term cardiovascular changes in pregnancies affected by HDP, suggesting a cardiovascular origin of the disease. Despite new insights into the pathophysiology of HDP, concepts of therapy remain unchanged and evidence for improved maternal and neonatal outcome by using antihypertensive agents is lacking.

**Methods and analysis** A prospective observational case–control study, including 100 women with HDP and 100 healthy controls, which will assess maternal haemodynamics using the USCOM 1A Monitor and Arteriograph along with cardiovascular markers (soluble fms-like kinase 1/placental-like growth factor, N-terminal pro-B type natriuretic peptide) in women with HDP under antihypertensive therapy, including a follow-up at 3 months and 1 year post partum, will be conducted over a 50-month period in Vienna. A prospective, longitudinal study of cardiovascular surrogate markers conducted in Oslo will serve as a comparative cohort for the Vienna cohort of haemodynamic parameters in pregnancy including a longer follow-up period of up to 3 years post partum. Each site will provide a dataset of a patient group and a control group and will be assessed for the outcome categories USCOM 1A measurements, Arteriograph measurements and Angiogenic marker measurements. To estimate the effect of antihypertensive therapy on outcome parameters, ORs with 95% CIs will be computed. Longitudinal changes of outcome parameters will be compared between normotensive and hypertensive pregnancies using mixed-effects models.

**Ethics and dissemination** Ethical approval has been granted to all participating centres. Results will be published in international peer-reviewed journals and will be presented at national and international conferences.

---

### STRENGTHS AND LIMITATIONS OF THIS STUDY

⇒ A strength of this study is that changes in maternal haemodynamics in combination with cardiovascular markers will be assessed prior to and after antihypertensive treatment in women with hypertensive disorders of pregnancy compared with a healthy gestational age-matched control group.

⇒ Another strength of the analysis is the comparison between single and combined oral antihypertensives.

⇒ A potential limitation of the study worth noting is that the comparison between different groups of antihypertensive agents might pose a challenge as the sample sizes within the respective groups will be small as we assume that a large proportion of patients will receive only one antihypertensive drug.

---

## BACKGROUND

Hypertensive disorders of pregnancy (HDP) are still the leading cause of maternal and perinatal morbidity and mortality worldwide.[1 2] Women undergo significant physiological systemic changes during pregnancy. Regarding the cardiovascular system, pregnancy acts as a cardiac stress model. Despite echocardiographically detectable changes, most pregnant women tolerate the inflicted cardiac stress without development of clinical symptoms.[3] HDP, in particular the pregnancy-specific syndrome pre-eclampsia, have been found to be common risk factors for future cardiovascular disease (CVD), even in previously healthy women without evidence of previous CVD.[4 5] The underlying pathophysiology and pathomechanism leading to pre-eclampsia are not yet fully understood and have been the focus of research in the last decades. Recently, the common hypothesis of

pre-eclampsia being a placental disorder was challenged as a number of studies show evidence for short-term and long-term cardiovascular changes in pregnancies affected by pre-eclampsia, possibly suggesting a cardiovascular origin of the disease. These cardiovascular changes include impaired cardiac output, an increase in systemic vascular resistance and left ventricular diastolic dysfunction especially in women affected by early-onset pre-eclampsia.[6–10] Studies on maternal haemodynamic changes in women with pre-eclampsia revealed persistent long-term cardiac alterations of up to 2 years after pre-eclampsia and an increased lifetime risk of essential hypertension and CVD.[11] Although there are new insights into a potential cardiovascular origin of the disease, the strategies of treating pre-eclampsia have not changed fundamentally in the last years. Furthermore, there is no international consensus on when and how increased blood pressure in pregnancy should be treated.[12] Studies of maternal haemodynamics in women suffering from HDP on antihypertensive treatment indicate effectiveness of haemodynamically guided antihypertensive therapy.[13] The data to support this, however, are limited. Khalil et al demonstrated significantly lower arterial stiffness in women treated with alpha-methyldopa, the first-line treatment of hypertension in pregnancy[14] in many countries. A study by Stott et al, evaluating antihypertensive therapy with labetalol and nifedipine guided by haemodynamic parameters in pregnant women with HDP, was able to demonstrate a significant decrease in episodes of severe hypertension when treatment was guided by haemodynamic parameters.[15] Studies on non-pregnant women undergoing haemodynamic monitoring while receiving antihypertensive therapy support this approach of a haemodynamically guided therapy when blood pressure is resistant or refractory.[12] Furthermore, a recent study by Mulder et al, assessing possible risk reduction of recurrent pre-eclampsia by introducing early tailored treatment of non-physiological haemodynamic changes during pregnancy in women with previous pre-eclampsia, showed a significant risk reduction of recurrent pre-eclampsia in this high-risk population.[16] Cardiovascular markers such as soluble fms-like kinase 1 (sFlt-1) and placental-like growth factor (PlGF) have been introduced into clinical practice to help estimate the time to delivery in women with pre-eclampsia.[17] A study by Sugulle et al indicated the usefulness of midregional proatrial natriuretic peptide as a biomarker in pre-eclampsia, both in humans and in a rat model, likely to reflect cardiovascular haemodynamic stress in these women.[18] N-terminal pro-B type natriuretic peptide (NT-proBNP) is known to be released by cardiomyocytes due to ischaemia, and increased mechanical strain seen in women with pre-eclampsia and also in non-pregnant patients with heart failure showed potential in predicting time to delivery in combination with the sFlt-1/PlGF ratio within 1 week.[19] In a prospective study assessing biophysical and biochemical markers of cardiovascular strain and placental dysfunction in patients with HDP and healthy controls, Verlohren et al

also demonstrated the additive predictive value of total peripheral resistance index, with additional moderate contribution of NT-proBNP and cardiac index in combination with sFlt-1/PlGF ratio for the prediction of HDP.[20]

### Objectives

This project aims to evaluate cardiovascular parameters in women affected by HDP by assessing maternal haemodynamic function during pregnancy as well as in the postpartum period up until 3 years after delivery. The focus of this study is to evaluate maternal haemodynamics as well as angiogenic and cardiovascular markers (sFlt-1/PlGF, NT-proBNP) under antihypertensive therapy in women affected by HDP compared with a healthy control group matched for gestational age, in order to assess whether cardiac indices should be used as a guiding tool for antihypertensive therapy in these women. In this study, we focus on changes in maternal haemodynamics secondary to antihypertensive treatment, also comparing single and combined oral antihypertensive therapy in women with HDP.

## METHODS
### Assessment of maternal haemodynamic function in pregnancy and post partum: Vienna

This is a prospective case–control study over a 50-month period, assessing women with HDP, including both gestational hypertension and pre-eclampsia, as well as a control group composed of healthy pregnant women matched for gestational age. All women will be recruited at the high-risk maternal outpatient clinic and antenatal ward at the Department of Obstetrics and Fetomaternal Medicine at the Medical University of Vienna. HDP, including gestational hypertension and pre-eclampsia, will be diagnosed according to the International Society for Hypertensive

---

**Box 1   Inclusion criteria**

Pregnant women 20+0 and 42+0 weeks of gestation:
⇒ With a singleton pregnancy.
⇒ Aged 18 years or older.
⇒ Able to give valid informed consent.
⇒ Presenting with gestational hypertension or pre-eclampsia.

---

**Box 2   Exclusion criteria**

⇒ Major fetal anomaly or aneuploidy/genetic syndrome.
⇒ Maternal and fetal cardiac defects (eg, congenital and non-congenital cardiac anomalies, heart failure, coronary heart disease).
⇒ Systemic cardiovascular diseases (eg, systemic lupus erythematosus with end-organ dysfunction).
⇒ Diabetes mellitus types I and II.
⇒ Chronic hypertension (predating pregnancy or onset <20 weeks of gestation).
⇒ Chronic kidney disease (eg, glomerulonephritis, polycystic kidney disease, renal insufficiency predating pregnancy).

---

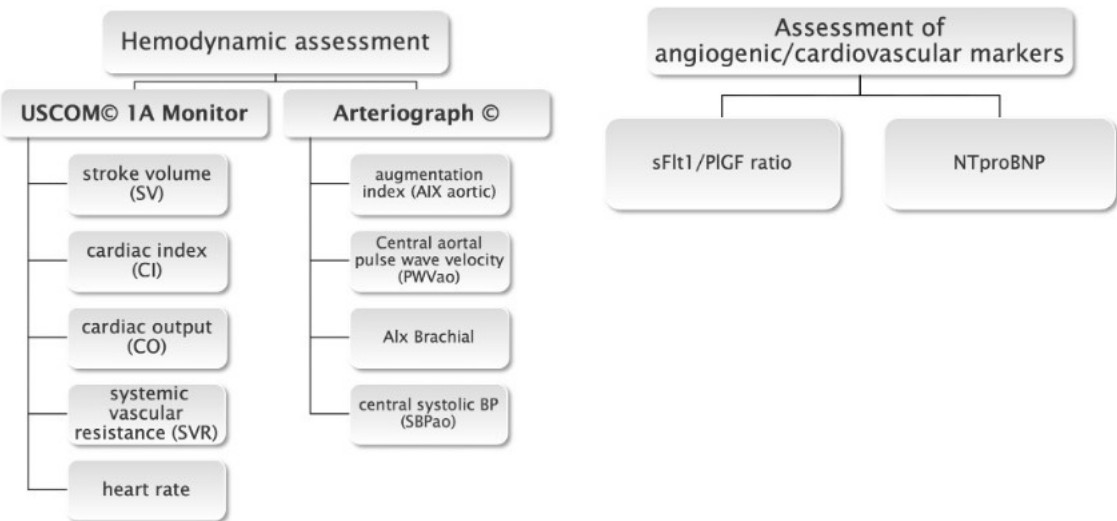

**Figure 1** Haemodynamic and laboratory assessments at each visit. BP, blood pressure; NT-proBNP, N-terminal pro-B type natriuretic peptide; PlGF, placental-like growth factor; sFlt-1, soluble fms-like kinase 1.

Disorders of Pregnancy 2021 revised criteria.[21] Gestational hypertension is defined as de novo systolic blood pressure of ≥140 mm Hg and/or diastolic blood pressure ≥90 mm Hg on two separate occasions ≥24 hours apart after 20 weeks of gestation. Pre-eclampsia is defined as de novo hypertension (≥140/90 mm Hg) after 20 weeks of gestation with the coexistence of proteinuria (protein/creatinine ratio ≥30 mg/mmol), other maternal organ dysfunctions including hepatic dysfunction (elevated transaminases, alanine aminotransferase or aspartate aminotransferase >40 IU/L) with or without right upper quadrant or epigastric pain, renal insufficiency (creatinine ≥100 µmol/L antenatally, ≥130 postnatally), neurological symptoms (eg, altered mental state, severe headaches, persistent visual scotoma, eclampsia), haematological complications (thrombocytopenia: platelet count <150 $10^9$/L, disseminated intravascular coagulation, haemolysis), pulmonary oedema or uteroplacental dysfunction (fetal growth restriction).[21]

Women fulfilling eligibility criteria (see boxes 1 and 2) will undergo non-invasive assessment of maternal haemodynamics including determination of heart rate, cardiac output, cardiac index, stroke volume and systemic vascular resistance using USCOM 1A Monitor, a non-invasive assessment of arterial stiffness using Arteriograph as well as measurement of sFlt-1/PlGF ratio and NT-proBNP by taking blood samples at each haemodynamic assessment (figure 1).

Primary and secondary outcome measures are summarised in boxes 3–5.

USCOM 1A Monitor is a device using continuous wave Doppler through a non-imaging probe in the suprasternal notch. It assesses velocity time integrals of transaortic blood flow at the left ventricular outflow tract, obtaining a complete haemodynamic profile. The Arteriograph device is a validated method of non-invasive assessment of arterial stiffness by measuring pulse wave velocity, aortic augmentation index and central systolic

blood pressure using a simple upper arm cuff. Serum sFlt-1/PlGF ratio and NT-proBNP will be measured using an automated immune analyser (Roche Diagnostics, Germany) at each haemodynamic assessment. Initial evaluation, if possible, will take place prior to starting antihypertensive treatment, followed by assessment after 24–72 hours after initiation of antihypertensive therapy. The same measurements will be repeated if therapy is extended by another antihypertensive agent. If no additional antihypertensive therapy is necessary, another assessment will be performed in the third trimester or prior to delivery if indicated due to critical maternal or neonatal situation. In the postpartum period, assessments will be undertaken within 1 week after delivery, and after a follow-up period of 3–6 months and 1 year (see figure 2). Antihypertensive treatment will be initiated as per local protocol, which is based on the updated Association of the Scientific Medical Societies in Germany guideline.[22] The control group will have serial haemodynamic assessment and blood sampling for cardiovascular parameters (as described for the cases group previously) in a 4-week interval during pregnancy and within a 1-week interval after birth, after 3–6 months and 1 year post partum.

## Sample size

The sample size estimation was done using the G*Power software (V.3.1.9.6, Universität Düsseldorf, Germany).[23] We estimated the sample size assuming a repeated measures analysis of covariance model to test the between-factor significance with the following parameters: effect size f: 0.15; error probability: 0.05, power: 0.90. We used a small to moderate generic effect size instead of tailoring it to a specific parameter, as a number of cardiovascular parameters will be compared between the groups. We further assumed an average of four measurements per group with a weak correlation between repeated measures (0.2). Under these assumptions, a total sample size of 220

---

**Box 3    Primary outcomes**

⇒ Comparison of maternal haemodynamics (figure 1) and angiogenic factors (soluble fms-like kinase 1/placental-like growth factor, N-terminal pro-B type natriuretic peptide) in women with hypertensive disorders of pregnancy (HDP) compared with healthy controls.

⇒ Assessment of maternal haemodynamics (figure 1) in women with hypertensive disorders of pregnancy under antihypertensive therapy:
   ⇒ Assessment of haemodynamic parameters before and after treatment.
   ⇒ Assessment of maternal haemodynamic parameters between women treated with different groups of antihypertensive agents.

⇒ Assessment of angiogenic factors in women with HDP under antihypertensive therapy:
   ⇒ Correlation of changes in haemodynamic parameters and angiogenic factors in women under antihypertensive treatment.

---

(110 patients in each) was estimated to reach the targeted power level.

### Postnatal longitudinal assessment of cardiovascular surrogate markers: Oslo

Oslo University Hospital, Ullevål will provide a comparative cohort for the Vienna cohort of haemodynamics in pregnancy. The Norwegian 'HAPPY' ('Health after pregnancy complications') Study, which currently comprises 221 women at 1 year and 108 women at 3-year follow-up, among others, including women after HDP and control patients, will be able to provide longitudinal data on cardiovascular surrogate markers such as endothelial function, carotid artery wall structure and aortic stiffness. Also, circulating biomarkers for CVD risk, such as systemic inflammatory marker high-sensitivity C reactive protein, and systemic 'metabolic stress' markers leptin, insulin resistance via Homeastasis Model Assessment index (HoMa index), fasting blood lipids (triglycerides, low-density lipoprotein (LDL)-cholesterol, high-density lipoprotein-cholesterol, oxidised LDL), measured by immunoassays and multiplex assays.[24] Additionally, the Oslo Pregnancy Biobank contains maternal blood and umbilical blood as well as placental tissue samples for

---

**Box 4    Secondary outcomes: maternal adverse outcome**

⇒ Acute renal insufficiency (defined as creatinine ≥100 µmol/L antenatally or ≥130 postnatally) or need for dialysis.

⇒ Placental abruption (retroplacental clot or associated with preterm delivery or fetal demise) described clinically.

⇒ Admission to the intensive care unit.

⇒ Days of hospital admission.

⇒ Mode of delivery.

⇒ Postpartum haemorrhage (defined as more than >500 mL blood loss after delivery).

⇒ Development of chronic hypertension in the follow-up period.

⇒ Hepatic dysfunction (elevated transaminases, alanine aminotransferase or aspartate aminotransferase >40 IU/L).

---

**Box 5    Secondary outcomes: perinatal/neonatal outcome**

⇒ Gestational age at delivery.

⇒ Birth weight.

⇒ Stillbirth.

⇒ Neonatal death.

⇒ Neonatal morbidity (admission to neonatal unit for more than 48 hours, respiratory distress syndrome, need for intubation, intraventricular haemorrhage, confirmed infection, necrotising enterocolitis, seizures, encephalopathy, retinopathy of prematurity).

⇒ Need for admission to neonatal intensive care unit (NICU).

⇒ Need for admission to special care baby unit.

⇒ Number of bed nights in NICU associated with delivery for preeclampsia.

---

assessment of morphological changes and will add additional comparative data.

Furthermore, the CHASE Study ('Cardiovascular health after pregnancy complications') will serve as another postpartum cohort, including blood samples and surrogate markers for endothelial function of 40 mother–child pairs 5–8 years post partum.[25]

### Timeline

The trial commenced in July 2019 and the planned end date is September 2023.

### Patient and public involvement

Patients or the public will not be involved in the design, conduct, reporting or dissemination plans of our research.

### Statistical analysis

Each site will provide a unique dataset comprising a patient group and a control group. The patient group will be divided into two subgroups: early-onset pre-eclampsia (onset before 34 weeks of gestation) and late-onset pre-eclampsia (onset after 34 weeks of gestation) for statistical analysis. Three sets of analyses are planned for each outcome category (USCOM measurements, Arteriograph measurements, Angiogenic marker measurements).

#### Baseline differences between hypertensive and normotensive pregnancies

Measurements at the time of enrolment will be used for this analysis. Variables will be described using location and scale parameters (mean, SD, median, IQR, etc) appropriate for their distribution characteristics. Groups will be compared after matching for predefined confounders (body mass index and gestational age at assessment). Mean differences or OR with 95% CIs will be reported to describe the effect sizes. Effect sizes will be estimated with generalised estimating equations using match identifiers as a cluster.

#### Estimating the effect of antihypertensive therapy on outcome parameters

Measurements at the time of enrolment and after the initiation of antihypertensive therapy (24–72 hours) will be used for this analysis. Paired measurements will be

---

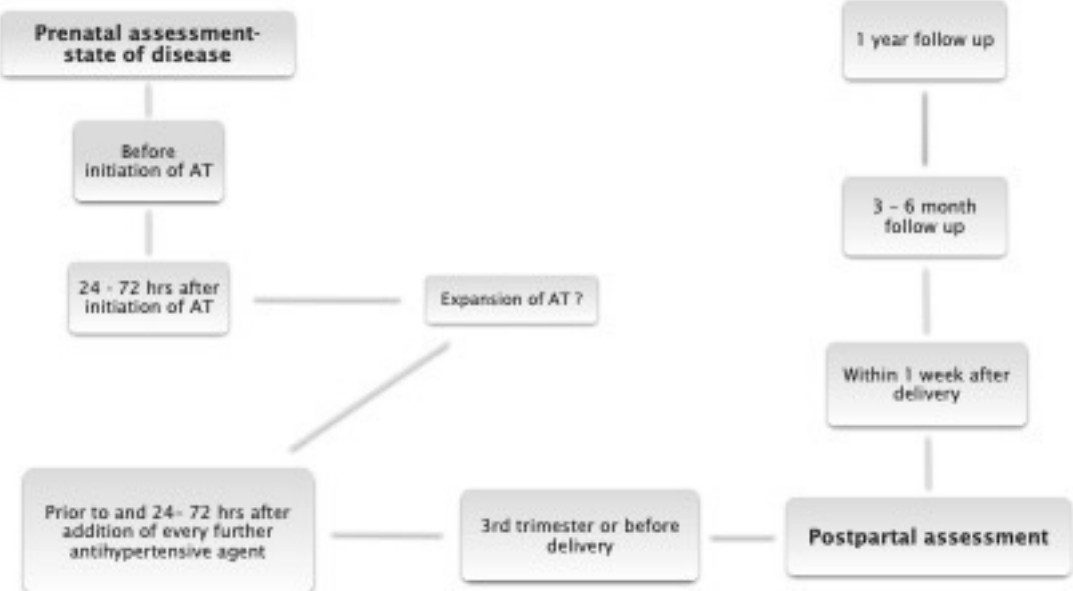

**Figure 2** Trial algorithm: timing of measurements within the patient cohort. AT, antihypertensive therapy.

used to estimate the mean differences or OR with 95% CIs. For variables with skewed distributions, median differences and non-parametric bootstrapped CIs will be reported. Also, clinical factors associated with before and after changes will be investigated and will be adjusted if necessary.

### Longitudinal change of outcome parameters between hypertensive and normotensive pregnancies

Repeat measurements from normotensive pregnancies and hypertensive pregnancies, which were not medicated, will be used for this analysis. A two-stage approach will be adopted for analysis. First, generalised linear mixed-effects models (GLMMs) with appropriate link function and random slopes for patient identifiers will be used to model the change of outcome variables with gestational age. Random effects from GLMMs will be extracted and the effect of hypertension status will be tested with linear models. All analyses will be conducted using R for Statistical Computing Software (R foundation, Vienna, Austria) performed by an appointed statistician.

### Data monitoring

Study participants will be anonymised by assigning a sequential patient number. All data will be entered, processed and stored anonymously in a case report form in the online data management program Clincase by the study investigators.

### Ethics and dissemination

This study will be conducted in accordance with the principles of Good Clinical Practice. The local ethics committee at each participating centre has reviewed the trial protocol and patient information as well as informed consent form and ethical approval has been granted. All women eligible for participation will be asked to give written informed consent prior to participation in the trial. Results will be published in peer-reviewed journals and disseminated at national and international conferences.

### DISCUSSION

HDP are still the leading cause of maternal and perinatal morbidity and mortality worldwide and therefore constitute a major socioeconomical burden. The prevalence varies from 2% to 8% in the industrialised world, and while the underlying pathophysiology is still largely unknown,[26] an underlying cardiovascular origin of the disease is suspected.[27] Novel therapeutic strategies to prevent long-term consequences of the disease are necessary. In this observational cohort study, maternal haemodynamic changes will be evaluated in women with HDP as well as in healthy control subjects to gain more insight into cardiovascular changes in women with HDP. Apart from gaining additional information on the underlying pathophysiology of pre-eclampsia, the effects of antihypertensive therapy on maternal haemodynamics in HDP will be investigated in women with gestational hypertension and pre-eclampsia. Recent studies investigating the effect of antihypertensive treatment on maternal haemodynamics in women suffering from HDP indicate that the outcome can be improved by haemodynamically guided antihypertensive therapy.[15] Sufficient data to support this, however, are lacking. Therefore, the goal of this study is to prove that maternal haemodynamic assessment is beneficial in women with HDP and leads to an overall improvement of maternal and neonatal outcomes. This study will influence the development of new strategies in the management of HDP and hopefully warrant implementation of haemodynamic assessment in women with HDP.

**Author affiliations**
¹Division of Obstetrics and Feto-Maternal Medicine, Hospital of the Medical University of Vienna, Vienna, Austria
²Experimental and Clinical Research Center, a cooperation between the Max Delbrück Center for Molecular Medicine in the Helmholtz Association and Charité Universitätsmedizin, Berlin, Germany
³Max-Delbrück Center for Molecular Medicine in the Helmholtz Association (MDC), Berlin, Germany
⁴Charité - Universitätsmedizin Berlin, corporate member of Freie Universität Berlin, Humboldt Universität zu Berlin, Experimental and Clinical Research Center, Berlin, Germany
⁵DZHK (German Centre for Cardiovascular Research), partner site Berlin, Berlin, Germany
⁶Division of Gynecology and Obstetrics, Oslo University Hospital, Oslo, Norway
⁷Faculty of Medicine, University of Oslo, Oslo, Norway
⁸Department of Obstetrics and Gynecology, Koc University Hospital, Istanbul, Turkey
⁹Fetal Medicine Unit, St George's Hospital, London, UK

**Contributors** JB is the general project coordinator and principal investigator of the project. MS and NH are project coordinators and principal investigators at their respective sites. JB, MS, NH, EK, AK and PP conceived and designed the study. JB, MS, NH, EK, AK and PP drafted the manuscript. All authors have critically read, revised and approved the final manuscript.

**Funding** This work is funded by an ERA-CVD Grant for Early Career Scientists through the Austrian, German and Norwegian Research fund (Austria: FWF-grant number I 4149-B, Germany: grant number 01KL1911, Norway: Research Council of Norway, project number 297333).

**Competing interests** None declared.

**Patient and public involvement** Patients and/or the public were not involved in the design, or conduct, or reporting, or dissemination plans of this research.

**Patient consent for publication** Not required.

**Provenance and peer review** Not commissioned; externally peer reviewed.

**ORCID iD**
Pilar Palmrich http://orcid.org/0000-0003-2449-2575

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
