## [Reviewer comments · BMJ Open]

ARTICLE DETAILS

TITLE (PROVISIONAL)	Maternal hemodynamics in Hypertensive Disorders of Pregnancy under antihypertensive therapy (HyperDiP): study protocol for a prospective observational case control study
AUTHORS	Palmrich, Pilar; Haase, Nadine; Sugulle, Meryam; Kalafat, E.; Khalil, Asma; Binder, Julia

VERSION 1 – REVIEW

REVIEWER	Mulders, Annemarie Erasmus Medical Center, Gynaecology and Obstetrics
REVIEW RETURNED	20-Nov-2022

GENERAL COMMENTS	1. Is the research question or study objective clearly defined? - No. The research question is described in the method section of the abstract. First, I would like to comment on the English narrative, which is insufficient quality and therefore very difficult to read. The title of this study suggests treatment during pregnancy whereas most follow-up time is during the postpartum period. Furthermore, from this manuscript it was not clear which groups (from the different cohorts (Vienna: case/control), Oslo and rats?) are going to be compared and for which parameters. And at which time-points during pregnancy study-visits are scheduled for the cases and controls. 2. Is the abstract accurate, balanced and complete? - No. The aim of the study is not clearly mentioned in the abstract. In the methods section of the manuscript three cohorts are mentioned, but this is missing from the abstract and the outcome parameters and statistical analysis is missing in the abstract. 3. Is the study design appropriate to answer the research question? - No. The research question is unclear and as such it is impossible to accurately evaluate the study design. However, following the description of the outcome parameters, the use of the cohorts, in- and exclusion criteria, and the limited explanation of the statistics used to compare the three cohorts, we believe this study design is inappropriate. 6. Are the outcomes clearly defined? - No. In box 3, the primary outcomes are posed as questions. This is incorrect. Outcome parameters should be clearly defined. Further, the outcome parameters do not involve the use of controls, which is uncommon as it is a case-control study. Lastly, the outcome parameters also do not involve the use of the control cohorts. Finally, we would like to mention that references for the inclusion/exclusion criteria and definitions are missing. For
--

	example the definition for preeclampsia is not stated. Further, it seems women with severe (early-onset) and milder forms of preeclampsia seem to be treated equally in the study. We do believe a distinction between the 2 disease entities should be made which at least should be mentioned in the statistical analysis section. Lastly, the reason for the inclusion-/exclusion criteria is not stated. It seems inappropriate with respect to the study aims to exclude the groups mentioned in exclusion criteria. PM: study aims are unclear. 7. If statistics are used are they appropriate and described fully? No. As mentioned before, we have some concerns regarding the statistical analysis. There is no description of how the three cohorts will be compared. And further, in the statistical analysis section, suddenly a preeclamptic rat-model is mentioned, which was not previously mentioned (in the methodology of the study). And differences between early and late inclusion are not addressed in the statistical analysis. 8. Are the references up-to-date and appropriate? - No. No definitions are provided for the outcome parameters and in-/exclusion criteria. 15. Is the standard of written English acceptable for publication? - No. Overall, it took a lot of time to read and really understand this manuscript. Also: - According to the instructions for publishing a study protocol in BMJ open: please add anticipated starting and ending dates of the study. - Most important reason for rejection, is that we believe there is a major flaw in study methodology. The study protocol is not described clearly enough. A prospective observational case-control study is an appropriate study design to assess maternal hemodynamics and to evaluate hemodynamic changes after anti-hypertensive therapy. However, it is unclear from this manuscript how. Although the research topic is relevant, the objective of this project is not answered by this study protocol.
--	---

REVIEWER	Cottrell, Jesse Marshall University, Obstetrics and Gynecology
REVIEW RETURNED	27-Dec-2022

GENERAL COMMENTS	Excellent and well designed study that addresses a gap in medical knowledge. I am very much looking forward to reading the results of your study. I would like to direct your attention to two more recent publications in this area as one study mentioned in the Background section is from 2017 (Stott et al). Preventing Recurrent Preeclampsia by Tailored Treatment of Nonphysiologic Hemodynamic Adjustments to Pregnancy, Mulder et al, 5 Apr 2021 https://doi.org/10.1161/HYPERTENSIONAHA.120.16502Hypertension. 2021;77:2045–2053
--

	The effect of impedance cardiography directed antihypertensive therapy on fetal growth restriction rates and perinatal mortality in women with chronic hypertension, Cottrell et al, Pregnancy Hypertens. 2022 Jun;28:123-127. doi: 10.1016/j.preghy.2022.03.006. Epub 2022 Mar 16
--	---

VERSION 1 – AUTHOR RESPONSE

Responses to the reviewers

REVIEWER #1

Reviewer 1, Comment 1: The research question is described in the method section of the abstract. First, I would like to comment on the English narrative, which is insufficient quality and therefore very difficult to read.

Response to Reviewer 1, Comment 1: The authors thank the reviewer for the comment. We have made some changes and hope that the manuscript is now written in satisfactory English.

Reviewer 1, Comment 2: The title of this study suggests treatment during pregnancy whereas most follow-up time is during the postpartum period. Furthermore, from this manuscript it was not clear which groups (from the different cohorts (Vienna: case/control), Oslo and rats?) are going to be compared and for which parameters. And at which time-points during pregnancy study-visits are scheduled for the cases and controls.

Response to Reviewer 1, Comment 2:

We appreciate the comments. Firstly, we would like to clarify concerns about the timepoints of study visits. The majority of assessments at the Vienna study site will cover the prenatal period. With regard to the Vienna study site, the time points of study visits in the HDP cohort are described on page 8, lines 180 to 187 as well as in figure 2, while the time points of study visits for the control cohort can be found in the same section on page 8, lines 188 to 192.

The Oslo cohort will serve as a postnatal cohort. Please find the time points of assessments on page 9, lines 209-210 and line 220. The postnatal Oslo cohort will provide longitudinal data on cardiovascular surrogate markers such as NTproBNP, as well as cardiovascular function parameters such as aortic stiffness of both women after HDP and healthy controls. These data will add additional data to the postpartum data assessed in the Vienna study cohort.

Lastly, the preclinical rat model will not be included in the study protocol and has been removed from the methods.

Reviewer 1, Comment 3: The aim of the study is not clearly mentioned in the abstract. In the methods section of the manuscript three cohorts are mentioned, but this is missing from the abstract and the outcome parameters and statistical analysis is missing in the abstract.

Response to Reviewer 1, Comment 3:

We thank the reviewer for the valuable comment. The cohort description has been adapted and corrected. An overview of the planned statistics has been added to the abstract (page 2, lines 48-54).

Reviewer 1, Comment 4: The research question is unclear and as such it is impossible to accurately evaluate the study design. However, following the description of the outcome parameters, the use of

the cohorts, in-and exclusion criteria, and the limited explanation of the statistics used to compare the three cohorts, we believe this study design is inappropriate.

Response to Reviewer 1, Comment 4: We appreciate the concerns of the reviewer. However, the study was funded by an international research grant (ERA-CVD Grant for Early Career Scientists through the Austrian, German and Norwegian Research fund) and, therefore, underwent extensive peer review. None of the reviewers raised concerns that the research question cannot be answered with this study design. Furthermore, a statistician was of course involved in planning of the study and it's design.

This is a prospective case control study involving women with HDP and healthy control subjects. The objective is to compare hemodynamic parameters (primary outcome cardiac output) in women with HDP and healthy controls (pages 5 to 8 as well as figure 1). Furthermore, cardiac output is evaluated and compared before and after initiation of antihypertensive therapy (first line alpha methyl dopa) in women with HDP (page 8, lines 180-183). The sample size was calculated to evaluate this particular research question (page 8, lines 195 to page 9, line 203) by a statistician.

We apologize for any misunderstandings and hope that the concerns about the study design not being appropriate could now be erased.

Reviewer 1, Comment 5: In box 3, the primary outcomes are posed as questions. This is incorrect. Outcome parameters should be clearly defined. Further, the outcome parameters do not involve the use of controls, which is uncommon as it is a case-control study. Lastly, the outcome parameters also do not involve the use of the control cohorts. Finally, we would like to mention that references for the inclusion/exclusion criteria and definitions are missing. For example the definition for preeclampsia is not stated. Further, it seems women with severe (early-onset) and milder forms of preeclampsia seem to be treated equally in the study. We do believe a distinction between the 2 disease entities should be made which at least should be mentioned in the statistical analysis section. Lastly, the reason for the inclusion-/exclusion criteria is not stated. It seems inappropriate with respect to the study aims to exclude the groups mentioned in exclusion criteria. PM: study aims are unclear.

Response to Reviewer 1, Comment 5:

We thank the reviewer for the important comments and apologize for unclear definitions. We have adapted the outcome parameters and hope to have addressed the concerns adequately (Box 3). For detailed outcome parameters see Figure 1.

Furthermore, definitions of inclusion/exclusion criteria have been added, see page 5-6, lines 140-152 and Box 2.

Further, we absolutely agree that a distinction between early onset and late onset preeclampsia is essential for the analysis and we have therefore added this to the statistical methods section (see page 10, line 230-233). However, according to the most recent guidelines, the distinction between "severe" and "non-severe" preeclampsia is not recommended, as stated in the 2021 revised ISSHP guidelines (*Magee LA, Brown MA, Hall DR, et al. The 2021 International Society for the Study of Hypertension in Pregnancy classification, diagnosis & management recommendations for international practice. Pregnancy Hypertens 2022; 27: 148-69.*).

We appreciate the reviewer's concerns regarding in/exclusion criteria. The primary aim of this project is to evaluate maternal hemodynamic changes in women affected by HDP by assessing maternal hemodynamic function during pregnancy as well as in the postpartum period. It is evident that women with preexisting conditions affecting the cardiovascular system (cardiac anomalies, chronic hypertension, systemic lupus erythematoses, etc) have a different initial cardiovascular situation than women who do not have preexisting cardiovascular conditions. Considering this, we believe that the chosen exclusion criteria are justifiable and hope that any uncertainties could be addressed appropriately.

Reviewer 1, Comment 6: As mentioned before, we have some concerns regarding the statistical

analysis. There is no description of how the three cohorts will be compared. And further, in the statistical analysis section, suddenly a preeclamptic rat-model is mentioned, which was not previously mentioned (in the methodology of the study). And differences between early and late inclusion are not addressed in the statistical analysis.

Response to Reviewer 1, Comment 6: Thank you for the comment. The issues have been addressed, the mention of the preeclamptic rat model has been removed and the description of the cohorts has been modified in the methods section of the manuscript (page 10, line 230-233).

Reviewer 1, Comment 7: No definitions are provided for the outcome parameters and in-/exclusion criteria.

Response to Reviewer 1, Comment 7: This has been adapted accordingly (page 5-6, lines 140-152 and Box 2)

Reviewer 1, Comment 8: Overall, it took a lot of time to read and really understand this manuscript.

Response to Reviewer 1, Comment 8: We hope that the English of the manuscript now meets acceptable standards.

Reviewer 1, Comment 9: According to the instructions for publishing a study protocol in BMJ open: please add anticipated starting and ending dates of the study.

Response to Reviewer 1, Comment 9: A timeline has been added according to the instructions (page 9, line 222-223).

Reviewer 1, Comment 10: Most important reason for rejection, is that we believe there is a major flaw in study methodology. The study protocol is not described clearly enough.

A prospective observational case-control study is an appropriate study design to assess maternal hemodynamics and to evaluate hemodynamic changes after anti-hypertensive therapy. However, it is unclear from this manuscript how. Although the research topic is relevant, the objective of this project is not answered by this study protocol.

Response to Reviewer 1, Comment 10. We appreciate the feedback and apologize for any uncertainties and misunderstandings concerning the methodology. We hope that the concerns about the study design could be resolved.

REVIEWER #2

Reviewer 2, Comment 1: Excellent and well designed study that addresses a gap in medical knowledge. I am very much looking forward to reading the results of your study.

I would like to direct your attention to two more recent publications in this area as one study mentioned in the Background section is from 2017 (Stott et al).

Preventing Recurrent Preeclampsia by Tailored Treatment of Nonphysiologic Hemodynamic Adjustments to Pregnancy, Mulder et al, 5 Apr 2021
<https://doi.org/10.1161/HYPERTENSIONAHA.120.16502>Hypertension. 2021;77:2045–2053

The effect of impedance cardiography directed antihypertensive therapy on fetal growth restriction rates and perinatal mortality in women with chronic hypertension, Cottrell et al, Pregnancy Hypertens. 2022 Jun;28:123-127. doi: 10.1016/j.preghy.2022.03.006. Epub 2022 Mar 16

Response to Reviewer 2:

The authors thank the reviewer for the encouraging feedback and for pointing out 2 recent publications, which we read with great interest. We have added them to our introduction and updated our reference section accordingly (page 4, lines 115; 124-127)